# Seasonal statistical-dynamical prediction of the North Atlantic Oscillation by probabilistic post-processing and its evaluation

André Düsterhus[1,2]

[1]Institute of Oceanography, Center for Earth System Research and Sustainability (CEN), Universität Hamburg, Germany
[2]ICARUS, Department of Geography, Maynooth University, Ireland

**Correspondence:** André Düsterhus (andre.duesterhus@mu.ie)

**Abstract.** Dynamical models of various centres have shown in recent years seasonal prediction skill of the North Atlantic Oscillation (NAO). By filtering the ensemble members on the basis of statistical predictors, known as subsampling, it is possible to achieve even higher prediction skill. In this study the aim is to design a generalisation of the subsampling approach and establish it as a post-processing procedure.

Instead of selecting discrete ensemble members for each year, as the subsampling approach does, the distributions of ensembles and statistical predictors are combined to create a probabilistic prediction of the winter NAO. By comparing the combined statistical-dynamical prediction with the predictions of its single components, it can be shown that it achieves similar results to the statistical prediction. At the same time it can be shown, that unlike the statistical prediction the combined prediction has less years where it performs worse than the dynamical prediction.

By applying the gained distributions to other meteorological variables, like geopotential height, precipitation and surface temperature it can be shown that evaluating prediction skill depends highly on the chosen metric. Besides the common anomaly correlation (ACC) this study also presents scores basing on the Earth Mover's Distance (EMD) and the Integrated Quadratic Distance (IQD), which are designed to evaluate skill of probabilistic predictions. It shows that by evaluating the predictions for each year separately compared to applying a metric on all years at the same time, like correlation based metrics, leads to

different interpretations of the analysis.

## 1 Introduction

Seasonal prediction of the North Atlantic Oscillation (NAO) is a challenge. During the year the NAO describes a high portion of the explained variability of the pressure field over the North Atlantic region and with it has a high influence on European

weather. While the Winter-NAO (WNAO) is a dominant factor for changes in the storm tracks over the North Atlantic (Hurrell, 1995), the Summer-NAO (SNAO) is associated with precipitation and temperature differences between Scandinavia and the Mediterranean (Folland et al., 2009).

Predicting the WNAO on the seasonal scale is a longstanding aim of the community (Doblas-Reyes et al., 2003; Müller et al., 2005; Scaife et al., 2014) and various current seasonal prediction systems have demonstrated limited significant correlation skill for the WNAO (Butler et al., 2016). Dobrynin et al. (2018) has shown that by combining statistical with dynamical prediction, a much higher significant corellation skill is achievable. This paper applies an ensemble subsampling algorithm, which bases selection of ensemble members on their closeness to statistical predictors. The selected ensembles are then used to create a new sub-selected ensemble mean, which has for the NAO index, but also for many other variables and regions, a better prediction skill than the ensemble mean of all ensemble members.

Statistical-dynamical predictions basing on different strategies are common in many fields in geoscience. Gleeson (1970) developed a framework for the dynamical evolution of statistical distributions in phase space with applications to meteorological fields. Vecchi et al. (2011) apply a combined statistical-dynamical approach by using an emulator basing on dynamical forecasts to create seasonal hurricane prediction. Roulston and Smith (2003) developed a "best member" concept, which use verification statistics to dress a dynamical ensemble prediction. Statistical post-processing procedures to enhance forecast skill by dynamical models are applied in various ways in atmospheric science (Williams et al., 2014). Especially Bayesian Model Averaging (Raftery et al., 2005), which creates weights for ensemble members basing on their performance in a training period has been well established.

The focus of this paper is to implement the subsampling algorithm as a probabilistic post-processing procedure, demonstrated for the seasonal prediction of the WNAO. In contrast to Dobrynin et al. (2018), which worked with deterministic ensemble members, it interprets ensemble members and the statistical predictors as values with uncertainties. The combination of statistical and dynamical model does not happen by selecting the ensemble members directly, but by combinations of probability density functions to create a new probabilistic forecast. This approach allows us to evaluate a prediction skill not only for a long time series, but for each individual year. We use for this two newly developed skill scores, the 1D-continuous-EMD and the 1D-continuous-IQD score, basing on the Earth Mover's Distance (EMD) and the Integrated Quadratic Distance (IQD). The WNAO has a severe influence on various meteorological fields over the European continent. Therefore, we also use the probabilistic information of the prediction to create a weighted mean of the ensemble members, which creates a better hindcast skill for important meteorological variables like surface temperature and precipitation.

## 2 Data and Model

To demonstrate the procedure we use the seasonal prediction system based on the MPI-ESM (Dobrynin et al., 2018) with a model resolution of T63/L95 (200 km / 1.875°, 95 vertical layers) in the atmosphere and T0.4/L40 (40 km / 0.4°, 40 vertical layers) in the ocean (also known as mixed resolution, MR). As described by Baehr et al. (2015), we initialise in each November between 1982 and 2017 a 30 ensemble member hindcast from an assimilation run based on assimilated reanalysis/observations in the atmospheric, oceanic and sea-ice component. As observational reference we use the ERA interim reanalysis (Dee et al., 2011).

For the observations and the hindcasts the NAO is calculated by an EOF analysis (Glowienka-Hense, 1990). For the WNAO we calculate the mean sea level pressure field for December, January and February and calculate the EOF of the North Atlantic sector limited by $20°-80°$N and $70°$W$-40°$E.

## 3   Methodology

### 3.1   Seasonal prediction of the WNAO

The seasonal prediction of the WNAO for the period of 1982 to 2017 is shown in Figure 1. Every dot represents one WNAO value of one ensemble member, which has also available the full meteorological and oceanographical fields during the associated winter period. These hindcast predictions for the WNAO have a large spread, covering the range of the observations given by the reanalysis, but do not give indication for a specific NAO value 2 to 4 month ahead. As a general skill measure the community applies correlation skills. Those measures have indicated in recent years significant hindcast skill for several

different prediction systems (Butler et al., 2016).

### 3.2   Statistical-dynamical prediction

Our approach will be applied to every single year independently. As an example we choose the year 2010, which shows an extreme negative WNAO value. The first step is to generate one probability density function (pdf) for each ensemble member prediction ($e_i$) of the WNAO value, which is generated by a 2000 member bootstrap of the EOF fields (Wang et al., 2014). In

the bootstrap the first EOF field is recalculated by resampling the mean sea level pressure fields from each year. To create from these predictions a pdf for all ensemble members ($\mathcal{E}$), mixture modelling (Schölzel and Hense, 2011) at discrete NAO index values is applied:

$$\mathcal{E}(v) = \sum_{i \in \mathcal{I}} e_i(v) \qquad (1)$$

Here, $v$ corresponds to each value of the discretised NAO values, and $\mathcal{I}$ to the indices of the ensemble members. The chosen

resolution for the discretised NAO values is 0.01 and after creating the sum of all single member pdfs, the overall pdf $\mathcal{E}$ is normalised. For 2010 the results are shown in Figure 2. As expected from Figure 1, the dynamical model prediction has a very broad pdf equating to a low signal.

    To sharpen the prediction we introduce literature backed physical statistical predictors. As predictors ($p_i$) we use those defined by Dobrynin et al. (2018): sea-surface temperature in the northern hemisphere, Arctic sea-ice volume, Siberian snow

cover and stratospheric temperature in 100 hPa. All predictors and their influence on the WNAO have been discussed in the paper. For the physical validity of a prediction the selection of the correct predictors is essential and have to be adapted to any newly analysed phenomena individually. Each predictor makes a prediction from the climatic state taken from the ERA interim reanalysis (Dee et al., 2011) before the initialisation of the dynamical model for a WNAO value in the following winter. For the predictors a normalised index over the hindcast period is calculated by forming the mean over the significantly correlated

areas between the physical field and the WNAO index. It has been shown by a real forecast test in Dobrynin et al. (2018) that this approach is usable also in cases where the predictor is only formed with past information instead of the whole hindcast period.

We treat the predictors $p_i$ like the ensemble members before and apply an empirical mixture modelling. For the year 2010 the results are shown in Figure 3. Due to the limited number of predictors compared to the ensemble members, and in the shown case also due to their alignment, the resulting statistical prediction pdf ($\mathcal{P}$) is much sharper than the dynamical model prediction.

To create a combined prediction the two pdfs ($\mathcal{E}$ and $\mathcal{P}$) are after normalisation multiplied at each of the discretised NAO values:

$$\mathcal{M}(v) = \mathcal{E}(v) \cdot \mathcal{P}(v). \tag{2}$$

After another normalisation the final combined prediction $\mathcal{M}$ creates the statistical-dynamical prediction for the seasonal NAO prediction in the specific year. The pdf of the observations ($\mathcal{O}$) are determined by the same bootstrapping mechanism as the one applied for the hindcasts. The result for the year 2010 is shown in Figure 4. The pdf of the combined prediction is close to the one of the statistical predictions, but shows differences where there is additional information from the dynamical model prediction. Therefore, the combined prediction shows a clearer signal than the dynamical model prediction, which does not give any indication of a specific NAO value at all.

### 3.3 NAO evaluation

To evaluate the performance of the three different predictions ($\mathcal{E}$, $\mathcal{P}$ and $\mathcal{M}$) and compare the predictions with the observation, we use two different scores, basing on the same formulation. The first bases on the Earth Mover's Distance (Rubner et al., 2001). The one-dimensional EMD (Düsterhus and Hense, 2012) can be derived by

$$D_{EMD}(f,g) = \frac{1}{n_b} \sum_{i=1}^{n_b} |F(v_i) - G(v_i)|, \tag{3}$$

where f and g are two pdfs and F and G the associated cdfs. $n_b$ describe in this case the number of discretised values $v_i$ of the CDFs.

The second is the Integrated Quadratic Distance (IQD), which is defined in its discrete formulation as (Thorarinsdottir et al., 2013)

$$D_{IQD}(f,g) = \frac{1}{n_b} \sum_{i=1}^{n_b} (F(v_i) - G(v_i))^2. \tag{4}$$

It is to mention that IQD is similar to the Continuous Ranked Probability Score (CRPS), but defined for non-deterministic observations. As a consequence, while CRPS needs to have a point observation, the IQD can take into account the full uncertainty distribution of an observation.

We define the scores for both metrics by comparing the pdfs of the model prediction ($\mathcal{M}$), the observations ($\mathcal{O}$) and the

climatology ($\mathcal{C}$). It is calculated for any prediction $\mathcal{A}$ by:

$$q(\mathcal{A}, \mathcal{O}) = 1 - \frac{D(\mathcal{A}, \mathcal{O})}{D(\mathcal{C}, \mathcal{O})}. \tag{5}$$

When $D$ is $D_{EMD}$ we call the score 1D-continuous-EMD-score, when we apply $D_{IQD}$ it is the 1D-continuous-IQD-score.

In case of a perfect prediction the score becomes 1, a model prediction equal to a climatology 0 and negative for a worse prediction than the climatology. Since the NAO index is normalised for mean and standard deviation, we use as climatology a standard normal distribution $\mathcal{N}(0,1)$. Important to note here is that Thorarinsdottir et al. (2013) compared the two metrics (EMD as *area validation metric*). While EMD is a metric measuring the distance between the pdfs it is in contrast to IQD not a proper divergence measure. As a consequence, the EMD prefers, unlike the IQD, underdispersed model simulations. In the following we will demonstrate the effect that the choice of the two different metrics have on the evaluation.

To estimate uncertainties, we use 500 randomly selected uniformly distributed weightings of the ensemble members between 1 and 0 and create with those a pdf for the scores.

### 3.4 Variable field evaluation

To estimate the post-processed variable field, we calculate a weighted mean of the meteorological variable fields, where the field of each individual member is weighted by a coefficient $c_i$. The weighting coefficients $c_i$ are estimated by weighting the predictions $\mathcal{A}$ (each of $\mathcal{E}$, $\mathcal{M}$ and $\mathcal{P}$) with each of the pdfs of the ensemble members ($e_i$):

$$c_{A,i} = \sum_v e_i(v) \cdot \mathcal{A}(v) \tag{6}$$

By weighting each ensemble member with its associated coefficient $c_{A,i}$ and calculating the weighted mean of the atmospheric fields of the individual ensembles then generates the model prediction for the specified field and prediction.

For evaluating the meteorological variable fields we apply three different strategies. The first is the Anomaly Correlation Coefficient (ACC), a common measure of skill in seasonal predictions. The second and third approach are using the 1D-continuous-EMD and 1D-continuous-IQD score at every grid point. As a climatology all observational values for the investigated time frame are chosen. The observation in each year is a single value with 100% as a weight. In case of the weights for the ensemble member, each value of the variable at the grid point gets weighted with the relative weight $c_{A,i}$ given by the three different prediction. With this approach it is possible to calculate the 1D-continuous-EMD and 1D-continuous-IQD scores for each of the three different predictions. In section 4.2.2 the relative positioning between two predictions is shown. Significances are here determined by DelSole and Tippett (2016), which determines the skill significances by comparisons to random walks.

## 4 Results

### 4.1 Evaluating the seasonal NAO prediction

In a next step we evaluate the yearly performance of the WNAO prediction of the three different predictions ($\mathcal{E}$, $\mathcal{P}$ and $\mathcal{M}$) with the 1D-continuous-EMD and 1D-continuous-IQD score. Figure 5 shows that the results of the combined ($\mathcal{M}$) and the

**Table 1.** Count of years of relative positioning of the three different predictions using the median of the 1D-continuous-EMD score.

| rank | EMD | | | IQD | | |
|:---:|:---:|:---:|:---:|:---:|:---:|:---:|
| | dynamical | statistical | combination | dynamical | statistical | combination |
| 1 | 5 | 17 | 14 | 13 | 16 | 7 |
| 2 | 5 | 10 | 21 | 7 | 12 | 17 |
| 3 | 26 | 9 | 1 | 16 | 8 | 12 |

statistical ($\mathcal{P}$) prediction are clearly better performing than the the dynamical model results ($\mathcal{E}$). In most years, the combined and the statistical prediction demonstrates skill for the 1D-continuous-EMD score compared to a climatological prediction over the whole uncertainty range. The dynamical model prediction has less variability over the years in skill than the other two predictions and only in a few years is able to reach the average skill of the combined prediction. The median and interquartile range of the summed up prediction skill for all evaluated years for the combined prediction ($0.39$ $[0.21; 0.60]$), is higher compared to the dynamical ($0.12$ $[0.01; 0.22]$) and statistical ($0.37$ $[0.17; 0.56]$) prediction. There is only one year (2003), with a strong discrepancy of the combined and statistical prediction and a clearly negative score. For the 1D-continuous-IQD the results are less clear. In this case the the median of the combined prediction is much closer to the dynamical prediction than the statistical prediction. Also the uncertainty range for the combined and statistical prediction increases relative to the dynamical prediction, which can be explained by their sharpness. To better evaluate the performance of each prediction with respect to the other predictions, we determine the relative ranking of the median of each prediction in each year for both scores. The rankings are counted for the whole hindcast period and the results displayed in Table 1. For the 1D-continuous-EMD score the dynamical prediction has only in a few years a better prediction skill than the other two predictions. In the majority of the years its prediction skill is lower than both other predictions. Looking at the best prediction for each year, the statistical and combined prediction is on equal terms. Nevertheless, the combined prediction is much more unlikely to be the worst of the three predictions in a year, while the statistical prediction takes much more often the third rank. These results show that the combined prediction is closer to the statistical rather than the dynamical prediction. In case the combined prediction is not the best one, it is in almost all cases better as one of the two. As such it offers a smoothing of the prediction skill, preventing many worse predictions. In case of the 1D-continuous-IQD score the result differs clearly. Here the dynamical prediction is much more competitive. It shares almost equally with the statistical prediction the first place, while the statistical prediction hardly changes its statistics of positions. As a consequence the combined prediction is much more often on the last place. Still, it is the prediction with the most middle places of the three predictions, stressing the argument that the combined prediction is a mixture of the two other.

## 4.2 Analysis of atmospheric variable fields

### 4.2.1 Climatological analysis

In the following we investigate three different atmospheric variable fields: surface temperature, total precipitation and 500 hPa geopotential height. In figure 6 the results are shown for the winter (DJF) season with the Anomaly Correlation Coefficient (ACC).

For the winter surface temperature, the main areas of significant hindcast skill of the combined prediction can be found over large parts of the North Atlantic and in a band reaching from Northern France to Eastern Europe, sparing North Scandinavia and the Mediterranean. These results are comparable to those shown by Dobrynin et al. (2018). Comparing it to the dynamical prediction shows that the main significant increase in skill can be found over Western Europe with a general non-significant increase over the whole continent. Also some significant increase of prediction skill can be found in the Labrador Sea, while a significant decrease is located over Greenland. The comparison to the statistical prediction shows only small differences. The areas shown as significant have to be assumed as random and an artefact of the bootstrapping approach.

The total precipitation has significant positive hindcast skill North of the British Isles, east of the Baltic Sea, in the Mediterranean and between the Canaries and the Azores. Compared to the dynamical prediction the area east of the Baltic sea and the Mediterranean has significantly increased skill, while again compared to the statistical prediction not much change is detectable. Finally for the geopotential height, the hindcast skill for the combined prediction is found in areas over Iberian peninsula, between the Canaries and the Azores and between the British Islands and Greenland. Compared to the dynamical prediction, some increase of hindcast skill can be found over southern Scandinavia and east of Greenland. In the comparison to the statistical prediction the combined prediction shows significantly lower hindcast skill at areas over Greenland and the British Isles. This can be explained by the conditioning of the statistical on the NAO directly, while the dynamic component of the statistical dynamical prediction decreases the skill in the main influence areas of the NAO.

The analysis shows that there exist changes between the dynamical and the combined prediction. Generally, the hindcast skill of the combined prediction is very close to the one of the statistical prediction.

### 4.2.2 Analysing single years

In a next step, the same atmospheric fields are compared with the 1D-continuous-EMD score (Fig. 7). To prevent influences of biases and trends, the data is gridpoint-wise normalised and de-trended. Again the analysis shows the relative positioning of two predictions. For the surface temperature in Winter the difference between the dynamical and combined prediction is only significant in small patches distributed over the North Atlantic. Generally, no clear patterns can be identified. Especially the large significant areas determined by the ACC before do not show any significance with this score. The significant area in the ACC over western Europe has some increased values in favour of the combined predictions, but is not significant. In the comparison between the statistical and the dynamical prediction the increases and decreases are consistent with what has been seen for the dynamical compared to the combined prediction. This consistency shows that the statistical model plays a

dominating role in the combination. Better hindcast skill for the combined prediction compared to the statistical prediction can be identified in the west of the Mediterranean.

For the total precipitation the only significant change is a stretch north of Scandinavia. Also for the other comparisons for this variable the changes are small and do not show a consistent pattern. This is different for the geopotential height, where large areas in the North-East Atlantic and north of Scandinavia are significantly better represented in the combined prediction rather than the dynamical prediction. Both areas are not identified in the equivalent comparison with the anomaly correlation. The comparison of the statistical prediction compared to the dynamical prediction shows very similar patterns. The last comparison shows that the combination has areas between the Canaries and the Azores where it is significantly higher, while in large areas of Western Europe it has consistently better skill, but does not show significantly better skill.

This analysis shows that the three predictions do not have in all cases a clear relative ranking towards each other. Generally the results are very patchy and apart from north of Scandinavia no consistency can be seen.

In case of the analysis of the 1D-continuous-IQD score (Fig. 8) the comparison between the statistical and the combined model shows, in terms of significant areas, comparable results to the one seen in Fig. 7. When the two predictions are compared to the dynamic prediction the latter performs much better with this score than with the 1D-continuous-EMD score. While the general pattern of the areas stays the same, the dynamic prediction is in most areas the best prediction. Comparing the combined and the statistical model shows remarkably similar results as the 1D-continuous-EMD score. All these results are consistent with the results we have seen in section 4.1 for the single time series.

## 5   Discussion and Conclusion

This paper shows a post-processing procedure, generalising the newly established subsampling procedure by Dobrynin et al. (2018). By selecting not only single ensemble members, but utilising their uncertainty ranges, a much better understanding on the reason for its success is possible. As seen in section 3.2 the better prediction skill for the NAO by the combination of the statistical and dynamical model compared to the unprocessed dynamical prediction results from the sharper prediction of the statistical prediction. As by construction the different statistical predictions are highly connected towards the target value, in this case NAO, the predictor driven predictions result in higher skill. Furthermore, advantages of using this post-processing approach compared to the pure subsampling is the availability of non-parametric uncertainties for the predictions and the possibility of weighting the different ensemble members for the analysis of variable fields with unequal weights. As such especially outliers can therefore be much better handled, without giving them too high of a weight within the analysis.

Compared to the statistical prediction, the combined prediction achieves similar results for the NAO prediction. In a three-way comparison together with the dynamical prediction we have shown that it is not generally showing more skill than the statistical prediction, but it observes less negative outliers in skill. Nevertheless, in case of the atmospheric variable predictions, the prediction basing on predictors is not entirely a statistical prediction. The construction of weighting the ensemble members, leads to a statistical-dynamical prediction as well, where the weight of the dynamical model is less pronounced. As such, the skill between the two dynamical-statistical predictions is more similar in this case than the NAO prediction itself. We have

seen that the two categories of scores show the hindcast skill of the different forecasts from a different perspective. The 1D-continuous-EMD and 1D-continuous-IQD score allow to effectively evaluate the skill of two probabilistic results, like observations and predictions. The scores have similar characteristics like the RMSE in cases of undetected trends, different variability of different forecasts or a bias. In case of this study it is noted that the combined prediction is sharper than the dyanamical prediction for each years prediction, but also varies more from year to year. Also compared to the correlation, the two presented scores can decompose the skill in a consistent way for every single year.

As each year is compared to the climatology, a value close to the climatology can have a huge influence by creating substantive negative scores. To prevent this, the application of other references, like uniform distributions over the whole measurement range, can be an appropriate alternative. Comparing the results of the 1D-continuous-EMD and 1D-continuous-IQD scores shows that the latter infers a much harder penalty for miss-predictions. While the EMD metric uses a linear distance measure, the IQD divergence increases the distance by the square in equation 4. A discussion and comparison of the properties of two measures have been done by Thorarinsdottir et al. (2013). In the practical implementation done in this paper we have seen that the IQD tends to prefer a non-informative prediction over a wrong sharp prediction, while the EMD is more tolerant to wrong prediction in order to achieve a better score.

By evaluating the skill on a yearly basis and taking a look at the relative positioning the approach allows for a paradigm change as also described by DelSole and Tippett (2016). By counting the years in which one prediction is better than another, a single outlier cannot drive the whole verification result as it can do for correlations or RMSE. It also answers a typical question in forecast verification in a much more appropriate way: How sure can we be that a single prediction is better than another? The evaluation procedure presented here is able to quantify this answer for non-parametric predictions.

The anomaly correlation (ACC) is well used in literature and its main disadvantage are parametric assumptions in the interpretation of its results. We have seen that there are considerable differences when all years are evaluated at the same time, like it is done in a correlation based score, or the evaluation bases on evaluating single years. Correlations can be misleading and show skill where there is not necessary a good argument for it as it is prone to outliers. These discussions are well known when correlation-like measures are compared with distance-like measures, like the RMSE. Further progress in the creation of appropriate skill evaluation is therefore necessary. It is noted, that while we show in this analysis only the results for the winter season, the results for the summer season are comparable.

The methodology and verification techniques shown in this analysis are widely applicable within predictions of many different phenomena. This is especially valid in case of non-parametric datasets like in the analysis of extremes. The statistical-dynamical approach as illustrated here delivers consistent improved results compared to one of its components. Seen as a post-processing step, it forms a useful step to condition predictions on a physical basis in order to reduce noise and intensify the signal. Using non-parametric approaches in the analysis offers a more appropriate path to verify predictions in general.

*Data availability.* All data are stored at the DKRZ in archive and can be made accessible upon request (https://www.dkrz.de/up).

*Author contributions.* A. D. developed the methodology, designed the study, interpreted the results and wrote the manuscript.

*Competing interests.* There are no competing interest by the author.

*Acknowledgements.* The author would like to thank Johanna Baehr and Mikhail Dobrynin for the fruitful discussions. AD was supported by the University Hamburg's Cluster of Excellence Integrated Climate System Analysis and Prediction (CliSAP). AD was also supported by A4 (Aigéin, Aeráid, agus athrú Atlantaigh), funded by the Marine Institute and the European Regional Development fund (grant: PBA/CC/18/01). The author would also like to thank three anonymous reviewers and editor Sebastian Lerch for very helpful comments on this paper. Model simulations were performed using the high-performance computer at the German Climate Computing Center (DKRZ).

270

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

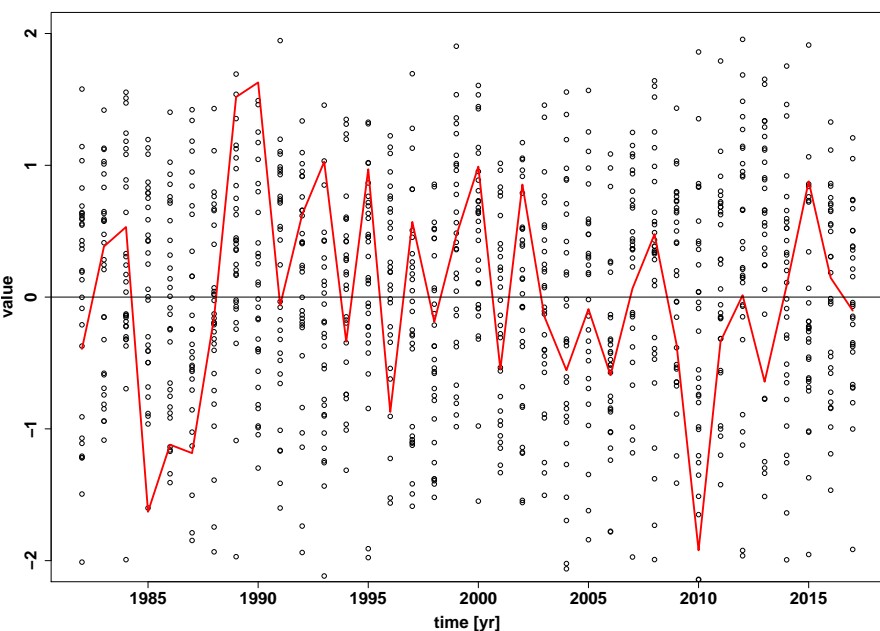

**Figure 1.** Seasonal prediction of the WNAO. Single dynamical models (black) initialised in November predicting the DJF-NAO (red).

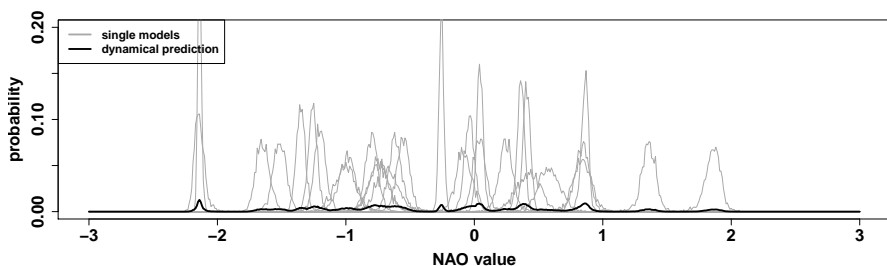

**Figure 2.** Dynamical prediction of the WNAO for 2010. Single models (grey) as pdf of their bootstrapped uncertainties. From this the overall model prediction (black) is created by empirical mixture modelling.

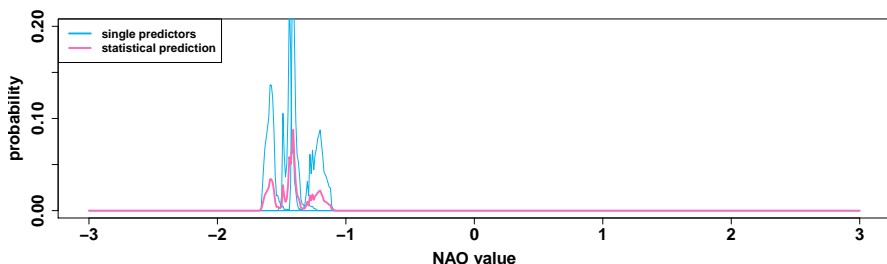

**Figure 3.** Statistical prediction of the WNAO for 2010. Single predictors (light blue) as pdf of their bootstrapped uncertainties. From this the statistical prediction (pink) is created by empirical mixture modelling.

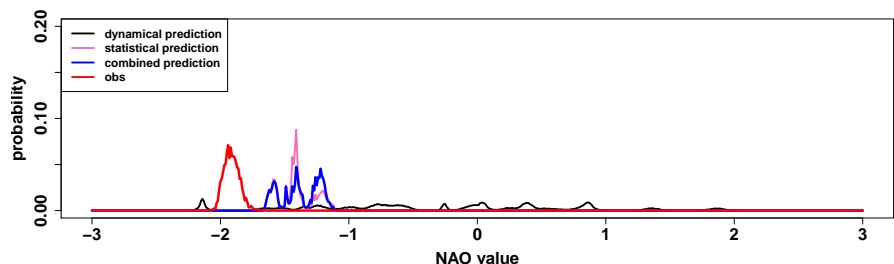

**Figure 4.** Sequence of post-processing procedure for the WNAO in 2010. Combining dynamical (black) and statistical (pink) to a combined prediction (blue) and comparing it to the observations (red).

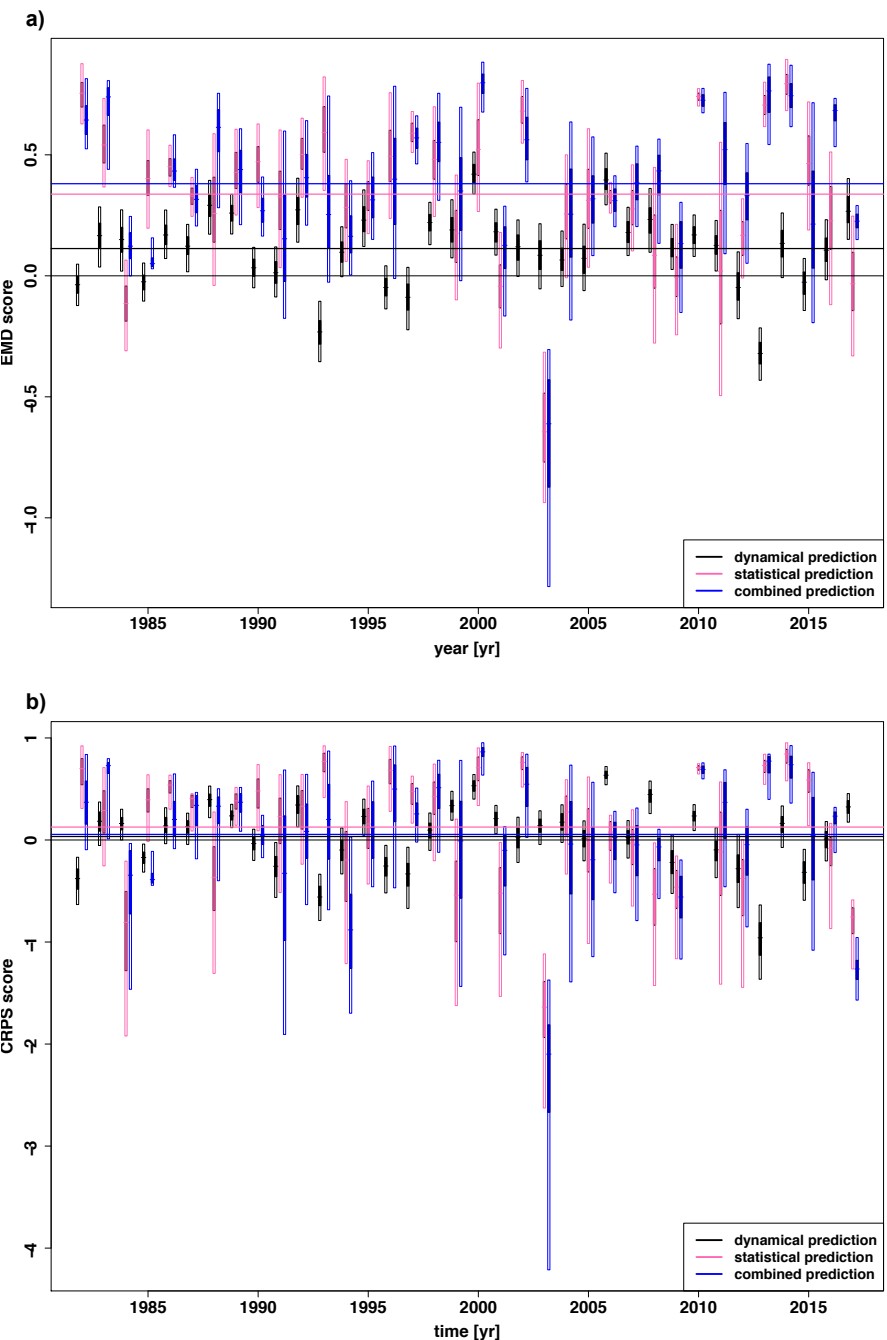

**Figure 5.** Yearly comparison of the WNAO scores for dynamical (black), statistical (pink) and combined prediction (blue). Each vertical bar represents the 5% to 95% bootstrapped 1D-continuous-EMD score (above) and 1D-continuous-IQD score (below). The filled part of these bars are the 25th to 75th quartiles and the small vertical lines the associated median. The long vertical lines are the averaged yearly scores for the different predictions.

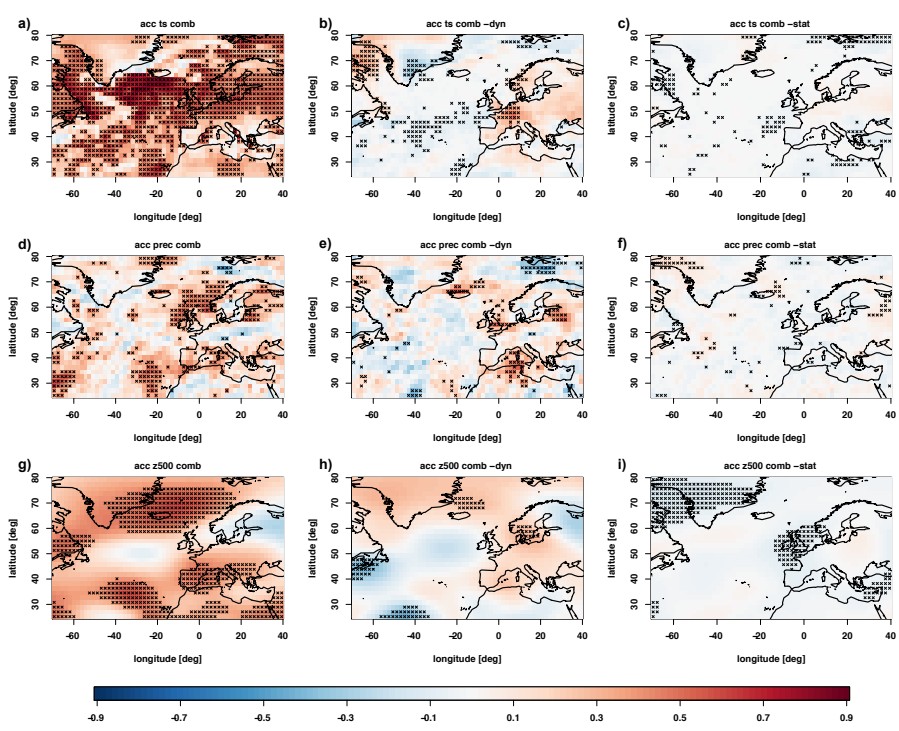

**Figure 6.** ACC results for the WNAO for three different atmospheric variables: surface temperature (first row), total precipitation (second row) and geopotential height (third row). Shown are the combined prediction (first column), the difference between the combined and the dynamical prediction (second column) and the difference between the combined and the statistical prediction (third column). Black dots indicate significances estimated by a 500 sample-bootstrap.

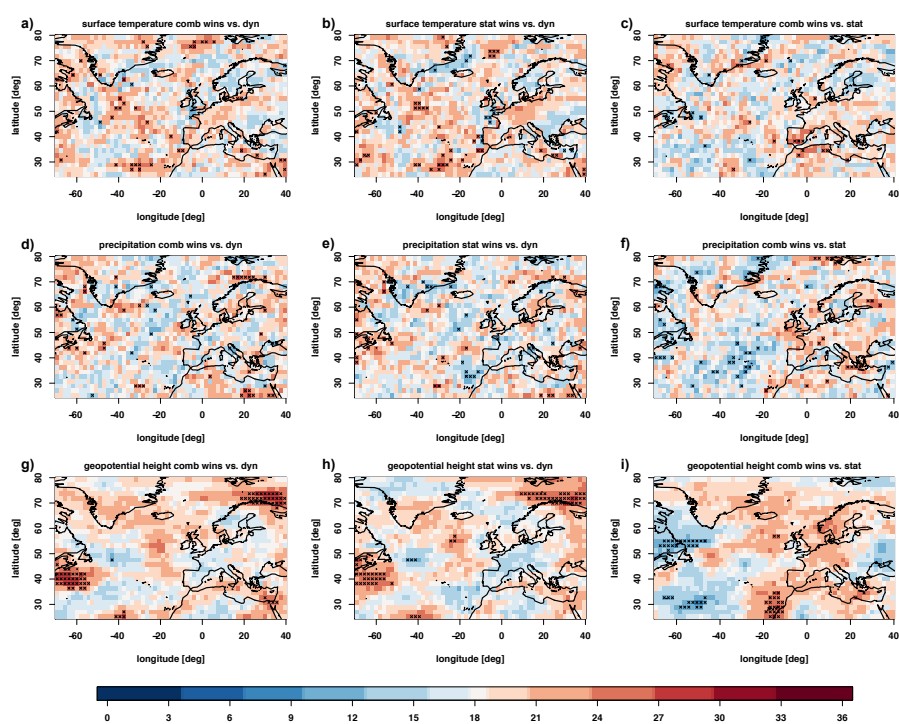

**Figure 7.** Relative positioning of the predictions of the variable fields on the basis of the 1D-continuous-EMD score. Shown is the number of years in which the first named prediction has a better score than the second named. Compared are the combined with the model prediction (first column), statistical with the dynamical prediction (second column) and combined with the statistical prediction (third column). Significances are determined by a comparison towards a random walk at a confidence level of 0.05. Variables are positioned as in figure 6.

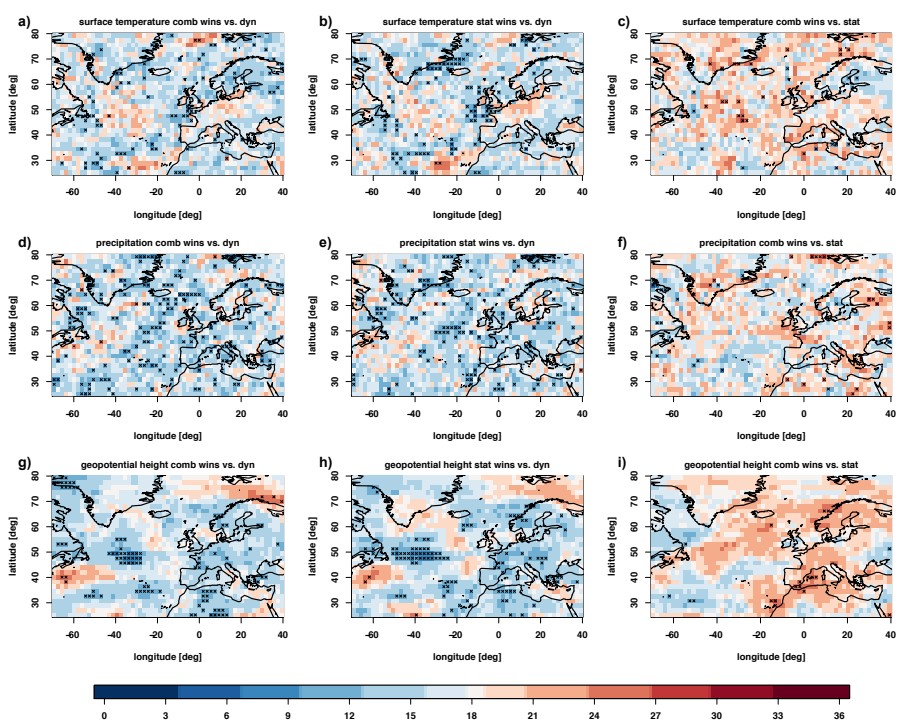

**Figure 8.** Relative positioning of the predictions of the variable fields on the basis of the 1D-continuous-IQD score. As 7 but calculated with the 1D-continuous-IQD.