# Peer review of "Seasonal statistical-dynamical prediction of the North Atlantic Oscillation by probabilistic post-processing and its evaluation"

_Nonlinear Processes in Geophysics, 2019_

## Referee Comment (RC1) · Anonymous Referee #1 · 21 Oct 2019

This study presents a novel empirical postprocessing method for combining dynamical seasonal weather forecasts with previously identified statistical predictors in order to achieve enhanced predictive skill. The proposed method involves constructing boot-strapped pdfs for individual ensemble member predictions. These are then combined and normalised to give a pdf. The statistical predictors are treated in the same manner. The normalised model and predictor pdfs are then combined by multiplication and normalised again to produce a final forecast pdf. Results for the NAO show improved performance of the combined pdf, particularly compared to the model pdf alone. Results for atmospheric fields are patchy, likely reflecting the teleconnections involved.

I congratulate the author on an interesting methodology and a concisely written manuscript. Although empirical in nature, the proposed post-processing method is clearly statistically motivated and attempts to capture and quantify relevant uncertainty. However, I would like to see a few details clarified, specifically:

Lines 56-60 – Please describe briefly the bootstrap procedure. It is not necessary to explain the block bootstrap proposed by Wang et al. (2014), but it is important to know what is being bootstrapped. I assume it is the daily pressure fields?

Section 3.3 - The Earth Mover's Distance is an interesting method for evaluating the quality of the forecasts, but in addition it would be useful to see an evaluation based on a conventional strictly proper scoring rule, such as the CRPS mentioned in the discussion.

Section 3.4 - I think "meteorological variable fields" in line 97 and "atmospheric field" in line 100 refers to the individual ensemble member fields? Please clarify.

Section 3.4 - I am unclear what exactly is taking place in Equation 5. Are these the $e_i(v)$ and $M(v)$ for the NAO forecast then used to weight entire fields for the atmospheric variables, or are these computed individually for each variable, or at each grid box for each variable? Please clarify.

Minor points and typos:

Lines 23-24: "The paper" -> "This paper", "bases on selection" -> "bases selection", "on the basis of statistical predictors" -> on "statistical predictors"

Line 33: "at the hand of" -> "for"

Section 2: Were the grid boxes latitude weighted before computing the EOFs?

Lines 66-71 - A table summarising the definitions of the predictors might be helpful here.

Line 81: "clearer signal as" -> "clearer signal than"

Lines 190-2: This doesn't make sense to me, sharper predictions should have less uncertainty. Do you mean variability between predictions? Please clarify

Lines 206-7: There are many good reasons not to use ACC (e.g., not a proper scoring rule), but the ACC does NOT assume normality. Some tests of ACC assume normality, but the statistic itself does not.
* * *

---

## Referee Comment (RC2) · Anonymous Referee #1 · 21 Oct 2019

The introduction would benefit from a paragraph placing the proposed method in context with similar existing methods such as Bayesian Model Averaging, see Williams et al. (2014) https://doi.org/10.1002/qj.2198 for a review.

---

## Referee Comment (RC3) · Anonymous Referee #2 · 4 Nov 2019

**1  General Comments**

In the study 'Seasonal statistical-dynamical prediction of the North Atlantic Oscillation by probabilistic post-processing' André Düsterhus proposes an approach for seasonal NAO prediction. This approach combines seasonal predictions of the MPI-ESM model with a statistical model based on meteorological fields as predictors. Here, for each ensemble member of the dynamical model a pdf is generated via bootstrapping. These single pdfs are merged to one pdf for all ensemble members via mixture modeling. The same procedure was applied to the statistical model, where each predictor of the statistical model was treated as one ensemble member. Subsequently the combined

model was constructed by multiplying the derived pdfs of the dynamical and the statistical model. In this study the author investigates the predictive performance of the combined model also w.r.t. to its single components, the dynamical model and the statistical model. For this purpose the anomaly correlation coefficient (ACC) and the Earth Mover's Distance (EMD) were applied. It can be shown that the performance of the combined model is superior to the dynamical model. On the other hand, an improvement w.r.t. the statistical model is not that clear. An analysis of the surface temperature, precipitation and geopotential height field shows significant skill of the combined model w.r.t. to ACC for many regions over Europa and North Atlantic. Moreover the combined model again outperforms the dynamical model but not the statistical model.

The general idea of the paper is interesting and the given approach is straight forward and certainly viable. This paper is well structured and written in a comprehensible language. The conlusions regarding the performance of the proposed statistical-dynamical prediction are justified by the results shown.

**2   Specific Comments**

**2.1   Title**

I would encourage you to reconsider the title, since I have the impression that a major part of the study is also about EMD-score and its benefits over the ACC.

**2.2   Introduction**

lines 36-38: In my opinion a short motivation is missing why you are also investigating other meteorological variables. For me, a link to the NAO-prediction and its evaluation

is missing within the introduction.

For a reader, who is not that familiar with subsampling it is hard to distinguish your work from the work of Dobrynin et al. (2018). So, I think that your introduction would benefit from a few sentences where you emphasize this aspect.

**2.3 Data and Model**

Maybe, one sentence about the used observational/reanalysis data would be helpful.

**2.4 Methodology**

lines 70-71: Please add some more details how the statistical model for the WNAO was constructed. Did you applied a linear regression model?
Moreover, please clarify whether you have applied a cross-validation. This is a very important aspect in the discussion about the prediction skill of the statistical model.
line 79: How is the pdf of the observations in Fig. 4 generated? Did you also applied a bootstrapping. Please clarify.
Eq. 3: What is $N$? I assume the number of time steps? Please use an other running index to avoid confusion with the ensemble number.
lines 107-108: What is the motivation for showing the relative positioning of the EMD-score? Isn't it also possible to show the relative positioning of the CRPS(S)? Did you calculate the EMD-score over every year as implied in Eqs. 3-4 or did you calculate the score for every year individually and derive the relative positioning afterwards? I assume the latter is valid but a more detailed description would prevent misunderstandings.

**2.5 Results**

lines 125-127: I don't understand that sentence. Do you mean that for years, where the combined prediction is not the best performing the model prediction performs best?
Figure 5 shows rather that the predictor predictions superior for such years...

**2.6 Discussion and Conclusion**

lines 182-184: Do you have any idea, why, on the one hand, the combined model is at first place almost as often as the statistical Model, but on the other hand is much more often at the second place? Is it possible, that the statistical model may not capture certain special cases, which are then taken into account by the dynamical model? Maybe an investigation of the corresponding years would certainly be very revealing.

**3   Technical Corrections**

line 17: the NAO describe → the NAO describes
line 19: (Hurrell (1995)) → (Hurrell, 1995)
line 21: (Folland et al. (2019)) → (Folland et al., 2019)
line 22: (Butler et al. (2016)) → (Butler et al., 2016)
line 32: sub-sampling → subsampling
line 85: (Rubner et al. (2001)) → (Rubner et al., 2001)
line 86: (Düsterhus and Hense (2012)) → (Düsterhus and Hense, 2012)
line 123: both other prediction → both other predictions

line 150: inflence → influence
line 170: three prediction → three predictions
line 200: (Thorarinsdottir et al. (2003)) → (Thorarinsdottir et al., 2003)
line 201: the approach allow → the approach allows
line 206: disadvantage ist → disadvantage is

———————————————————

---

## Referee Comment (RC4) · Anonymous Referee #3 · 14 Nov 2019

I agree with the general comments of both Reviewer 1 and Reviewer 2: the methodology is relatively straightforward and the text is well structured and easy to follow. While I found the use of the Earth Mover's Distance interesting, I support Reviewer 1 comments on using the CRPS. The reason for this being that if one is to introduce a new methodology, it is easier to demonstrate its usefulness by using the tools of the trade that people are already familiar with. Using such score will make it easier to compare with other post-processing attempts and might improve the chances that the methodology will be adopted.

Also, I found the terminology used to refer to the different forecast quite confusing. For

one, the author uses model to refer to the output of the dynamical forecast whereas really the 3 sets of predictions are actually models, albeit just different types of models. I would replace "model" with either dynamical model or the name of the forecast system (or something along those lines) throughout the text and in table 1. I would also suggest changing the name predictor model to something else, just to avoid the repetition with the word prediction (which comes up a lot) and to differentiate with the actual predictors (e.g. on line 147: "to the predictor prediction"). Maybe it can be changed to statistically-based, or something along those lines. And not refer to the combined model as a hybrid model or statistical-dynamical model, as in the introduction? Although referring to it as the combined model is ok and doesn't lead to confusion in this case.

**Introduction**

I think the manuscript would benefit from a short paragraph on the skill of climate models at predicting the NAO. At the moment, there is only 1/2 a sentence on this. In particular, they should mention that Scaife et al. (2014) do find skill at predicting the winter NAO with a different forecast system.

**Methodology**

I would mention the equivalent resolution of the atmosphere and ocean model in either km or degree. I would guess many readers won't know what T63 and/or T0.4 correspond to. I would also specify the components used to initialize the model as well as the observational dataset used to verify the predictions.

**Section 3.4**

It took me a while to understand what was meant by "To estimate the post-processed variable field", as all the discussion up until that point was about computing the NAO. I would add an introductory sentence to help the reader transition from the NAO to the meteorological variables we actually care about (tas, pr, ...).

And I'm with Referee 1 on equation 5: I don't quite understand what is happening here.

Minor points

Line 31: an dynamical -> a dynamical

Line 52-54: "The skill is ..." I don't understand this sentence. Please reformulate

Line 70: " Each predictor predicts statistically..." That's a bit awkward. You should reformulate.

Line 81" a clearer signal as -> a clearer signal than

Line 106: By this approach -> With this approach

Line 120: in contrast towards -> with respect to

Line 126: "but in case one of the two is better than the combined prediction, the latter beats in nearly all cases the one remaining". Isn't this what we would expect? If we are to average two models in a 3rd one and #1 is the best, I would expect that #3 would be #2.

Line 140: "are hardly existing". Please reformulate.

Line 143: "Towards the model prediction" I don't know what you mean here.

Line 148: In comparison to the predictor prediction... significantly decreased in their hindcast skill." Please reformulate this sentence.

Line 159: "In the comparison between.. can be explained by the statistical model." I don't understand what the author is trying to say here.

Line 181: "a too high weight -> too high of a weight

Line 187: "We have seen that ... in different ways" Do you mean that they provide a different perspective?

Line 193: "Also, unlike ... in a consistent way". Please rephrase.

Line 196: "As each year... To prevent this other references," I don't understand this.

Line 203: cannot anymore drive -> cannot drive

Line 205: The used evaluation -> The evaluation

Figures

I would combined Figures 2, 3 and 4 into Figure 2a, b, c. Also, I would put the x-axis range from -2.5 to 2.5.

The legend in most figure is quite difficult to read.

Figures 2-5: I would remove the [ ] when there are not units.

Figure 2: I would make the individual members more visible.

Figure 4: I would display the obs in black, as it is the reference.

Figure 5: I see no dark green in the figure, despite what the caption says. Also, I would pick different colors to differentiate between the different models: light blue and dark blue makes it difficult to see what's going on.

Figure 6: The font on the colorbar is way too small. Also, I would change the range of the color for columns 2 and 3.

Figure 7: The font on the colorbar is also way too small.

Scaife, A. A., et al. (2014), Skillful longrange prediction of European and North American winters, Geophys. Res. Lett., 41, 2514–2519, doi:10.1002/2014GL059637.

---

## Referee Comment (RC5) · Anonymous Referee #3 · 14 Nov 2019

I would also change the colorbar for Figure 7 to a monochrome as opposed to red/blue. Red/blue is better when we have 0 in mid-range, but it is not the case here.

---

## Author Comment (AC1) · 17 Dec 2019

**1   General**

I would like to thank all reviewers for the valuable comments. I have reframed the manuscript to incorporate the comments of all reviewers. Important things that have changed are the renaming of the predictions throughout the manuscript (now statistical, dynamical and combined), and the inclusion of the CRPS-like IQD as a new score. I have chosen to present the results of both, the EMD and the IQD, as I think it is important to show other scientists the direct effects that choices about scores, whether

they are proper or asymptotically proper, have and discuss the differences at hand of the appropriate literature. This added to the figures, which were reworked as well as debugged as some figures had problems in their used colourbars. Furthermore, I have corrected mistakes throughout the manuscript when I became aware of them. A detailed reply to each individual comment can be found below.

**2   Reviewer 1**

**This study presents a novel empirical postprocessing method for combining dynamical seasonal weather forecasts with previously identified statistical predictors in order to achieve enhanced predictive skill. The proposed method involves constructing bootstrapped pdfs for individual ensemble member predictions.   These are then combined and normalised to give a pdf.   The statistical predictors are treated in the same manner.   The normalised model and predictor pdfs are then combined by multiplication and normalised again to produce a final forecast pdf. Results for the NAO show improved performance of the combined pdf, particularly compared to the model pdf alone. Results for atmospheric fields are patchy, likely reflecting the teleconnections involved.**
**I congratulate the author on an interesting methodology and a concisely written manuscript.   Although empirical in nature, the proposed post-processing method is clearly statistically motivated and attempts to capture and quantify relevant uncertainty.**

I would like to thank you very much for these kind words.

**However, I would like to see a few details clarified, specifically:**
**Lines 56-60 - Please describe briefly the bootstrap procedure. It is not necessary**

[Figure]

**to explain the block bootstrap proposed by Wang et al. (2014), but it is important to know what is being bootstrapped. I assume it is the daily pressure fields?**

It will be clarified as:
"In the bootstrap the first EOF field is recalculated by resampling the mean sea level pressure fields from each year."
So no, I do not touch the daily fields, I work on a yearly basis.

**Section 3.3 - The Earth Mover's Distance is an interesting method for evaluating the quality of the forecasts, but in addition it would be useful to see an evaluation based on a conventional strictly proper scoring rule, such as the CRPS mentioned in the discussion.**

I went in the last weeks through my code and implemented the "CRPS". The consequences are a bit more substantial as I expected. So to the background: EMD is asymptotically proper, the CRPS-equivalent is k-proper. See for this Thorarinsdottir et al. 2013, where the "CRPS" is described as "Integrated quadratic distance" and EMD as "Area validation metric". To take account of these differences I include both metrics into the manuscript. I focus on the EMD in the explanations, as it is simpler to understand for non-statisticians. IQD is then discussed as a second case in a comparison to the EMD.

**Section 3.4 - I think "meteorological variable fields" in line 97 and "atmospheric field" in line 100 refers to the individual ensemble member fields? Please clarify.**

We have replaced the sentences with:
1. see the following point

2."By weighting each ensemble member with its associated coefficient $c_i$ and calculating the weighted mean of the atmospheric fields of the individual ensembles then generates the model prediction for the specified field."

**Section 3.4 - I am unclear what exactly is taking place in Equation 5. Are these the ei(v) and M(v) for the NAO forecast then used to weight entire fields for the atmospheric variables, or are these computed individually for each variable, or at each grid box for each variable? Please clarify.**

For clarification:
"To estimate the post-processed variable field, we calculate a weighted mean of the meteorological variable fields, where the field of each individual member is weighted by a coefficient $c_i$. The weighting coefficients $c_i$ are estimated by weighting the predictions $\mathcal{A}$ (each of $\mathcal{E}$, $\mathcal{M}$ and $\mathcal{P}$) with each of the pdfs of the ensemble members $(e_i)$"

Also the following equation is adapted. So technically I simply weight the whole ensemble member (or better all its fields) by the calculated coefficient and calculate then later an weighted ensemble mean.

**Minor points and typos:**
**Lines 23-24: "The paper" -> "This paper", "bases on selection" -> "bases selection", "on the basis of statistical predictors" -> on "statistical predictors"**
**Line 33: "at the hand of" -> "for"**

Corrected.

[Figure]

**Section 2: Were the grid boxes latitude weighted before computing the EOFs?**

Yes. The EOF are calculated with CDO and its user guide states that they are area weighted.

**Lines 66-71 - A table summarising the definitions of the predictors might be helpful here.**

On the suggestion of the other reviewer, I have stated more details on the predictors in this section. A table would be too much, as the paper is basing on the assumption that the statistical predictions as well as the dynamical one is given, and from that point the analysis starts.

**Line 81: "clearer signal as" -> "clearer signal than"**

Corrected.

**Lines 190-2: This doesn't make sense to me, sharper predictions should have less uncertainty. Do you mean variability between predictions? Please clarify**

Clarification:
"In case of this study it is noted that the combined prediction is sharper than the model prediction for each years prediction, but also varies more from year to year."

**Lines 206-7: There are many good reasons not to use ACC (e.g., not a proper scoring rule), but the ACC does NOT assume normality. Some tests of ACC**

[Figure]

**assume normality, but the statistic itself does not.**

I fear here is a misunderstanding. So ACC is in its core a correlation and a correlation bases in its basic assumptions that the two variables follow a bivariate normal distributed when the results interpreted in a standard way. So you are right that to calculate them and getting a result we do not need the normal distribution, but when we interpret the results, as we usually do, we assume the normal distribution indirectly. I had originally planned for this paper also to replace the ACC with a non-parametric version, but as the metric for this step is currently stuck in review, I had to take it out. To clarify that here and taking the discussion point out of this paper, I reformulate the sentence:
"The anomaly correlation (ACC) is well used in literature and its main disadvantage are parametric assumptions in the interpretation of its results."

**The introduction would benefit from a paragraph placing the proposed method in context with similar existing methods such as Bayesian Model Averaging, see Williams et al. (2014) https://doi.org/10.1002/qj.2198 for a review.**

I have added a couple of sentences to the introduction, so that beside dynamical-statistical predictions also post-processing procedures are covered.

**3   Reviewer 2**

**1 General Comments**
**In the study 'Seasonal statistical-dynamical prediction of the North Atlantic Oscillation by probabilistic post-processing' André Düsterhus proposes an**

[Figure]

approach for seasonal NAO prediction. This approach combines seasonal pre-
dictions of the MPI-ESM model with a statistical model based on meteorological
fields as predictors. Here, for each ensemble member of the dynamical model
a pdf is generated via bootstrapping. These single pdfs are merged to one
pdf for all ensemble members via mixture modeling. The same procedure was
applied to the statistical model, where each predictor of the statistical model
was treated as one ensemble member. Subsequently the combined model was
constructed by multiplying the derived pdfs of the dynamical and the statistical
model. In this study the author investigates the predictive performance of the
combined model also w.r.t. to its single components, the dynamical model and
the statistical model. For this purpose the anomaly correlation coefficient (ACC)
and the Earth Mover's Distance (EMD) were applied. It can be shown that the
performance of the combined model is superior to the dynamical model. On
the other hand, an improvement w.r.t. the statistical model is not that clear.
An analysis of the surface temperature, precipitation and geopotential height
field shows significant skill of the combined model w.r.t. to ACC for many
regions over Europa and North Atlantic. Moreover the combined model again
outperforms the dynamical model but not the statistical model.
The general idea of the paper is interesting and the given approach is straight
forward and certainly viable. This paper is well structured and written in a
comprehensible language. The conlusions regarding the performance of the
proposed statistical-dynamical prediction are justified by the results shown.

I would like to thank you very much for these kind words.

**2 Specific Comments**
**2.1 Title**
**I would encourage you to reconsider the title, since I have the impression that a**

[Figure]

**major part of the study is also about EMD-score and its benefits over the ACC.**

I am a bit hesitant to claim huge advantages compared to the ACC, it is simply another view and therefore complementary. Unfortunately the title is already quite long, but I have obliged with your request. Therefore, I have added "and its evaluation" to it. In the end it is a special issue on post-processing and so the post-processing should be in focus (as it is in the paper after all).

**2.2 Introduction lines 36-38: In my opinion a short motivation is missing why you are also investigating other meteorological variables. For me, a link to the NAO-prediction and its evaluation is missing within the introduction.**

Clarification:
"The WNAO has a severe influence on various meteorological fields over the European continent. Therefore, we also use the probabilistic information of the prediction to create a weighted mean of the ensemble members, which creates a better hindcast skill for important meteorological variables like surface temperature and precipitation."

**For a reader, who is not that familiar with subsampling it is hard to distinguish your work from the work of Dobrynin et al. (2018). So, I think that your introduction would benefit from a few sentences where you emphasize this aspect.**

The difference is in itself not huge, as the aim of the paper is to generalise the approach by Dobrynin et al 2018 and explain its results in a statistical rigorous way. I have added a clear statement where this manuscript deviates from Dobrynin et al. 2018, by adding:
"In contrast to Dobrynin et al. (2018), which worked with deterministic ensemble

members,..."
to the introduction of the statistical model. Also I added that the combination of statistical and dynamical model does not happen by selection, but by statistical combination of pdf.

**2.3 Data and Model Maybe, one sentence about the used observational/reanalysis data would be helpful.**

Clarification:
"As observational reference we use the ERA interim reanalysis (Dee et al 2011)."

**2.4 Methodology lines 70-71: Please add some more details how the statistical model for the WNAO was constructed. Did you applied a linear regression model?**

Clarification:
"For the predictors a normalised index over the hindcast period is calculated by forming the mean over the significantly correlated areas between the physical field and the WNAO index. It has been shown by a real forecast test in Dobrynin et al. 2018 that this approach is usable also in cases where the predictor is only formed with past information instead of the whole hindcast period."
As we focus in this paper on the explanation of how the method works and not anymore on how good it works, we do not re-perform here the real-forecast test. This is also due to the enormous amount of computational resources it would require to calculate it.

**Moreover, please clarify whether you have applied a cross-validation. This is a very important aspect in the discussion about the prediction skill of the**

[Figure]

**statistical model.**

The study does not include the jackknife (often called cross-validation, but technically that is a wrong term). Concerning the validity of the statistical model Dobrynin et al 2018 does extensive discussions on it, including applications of the jackknife in various ways. Applying a jackknife here is a bit critical, as the temporal variation is already used in the creation of the model by the bootstrapping of the EOF fields. Therefore the main uncertainty estimate is done over the 500 weightings of the ensemble members. Taking additionally single years out is in my view not sufficient to estimate the uncertainty of the pdfs. Furthermore, the required computational resources would be enormous. For significances in the ACC fields a bootstrapping is applied, as a jackknife generally underestimates the uncertainties and therefore has to be seen as a far worse test as the application of bootstrapping.

**line 79: How is the pdf of the observations in Fig. 4 generated? Did you also applied a bootstrapping. Please clarify.**

Clarification:
"The pdf of the observations ($\mathcal{O}$) are determined by the same bootstrapping mechanism as the one applied for the hindcasts."

**Eq. 3: What is N? I assume the number of time steps? Please use an other running index to avoid confusion with the ensemble number.**

It describes in this case the number of discretised values of the CDFs. It was originally defined for histograms, where it defines the number of bars. I have changed it to $n_b$.

[Figure]

**lines 107-108: What is the motivation for showing the relative positioning of the EMD-score? Isn't it also possible to show the relative positioning of the CRPS(S)? Did you calculate the EMD-score over every year as implied in Eqs. 3-4 or did you calculate the score for every year individually and derive the relative positioning afterwards? I assume the latter is valid but a more detailed description would prevent misunderstandings.**

CRPS has nowadays the problem that it is only been defined for evaluation towards deterministic observations. This general understanding of a CRPS in this way is highly problematic and leads often to many misunderstandings. By suggestion of Reviewer 1 I have taken now both scores into the paper with explanations and figures so show differences and consequences of the choices. Due to the clear definitions in the literature I have sticked to the naming as IQD and explained it in detail. The reason for preference for the EMD is generally its much simpler understandability for the common users, as it describes a real metric. Unfortunately, as I described in the manuscript it is just asymptotically proper. As I think it is important that such differences are not just buried in statistical literature, but are shown in real applications I show them here both. To make clear: it is not just about an introduction of a new score (where IQD is preferred), but also for explaining the way why subsampling (or its generalised version in this paper) works at all, where a simpler metric is certainly prefered (so therefore the inclusion of the EMD).

**2.5 Results lines 125-127: I don't understand that sentence. Do you mean that for years, where the combined prediction is not the best performing the model prediction performs best? Figure 5 shows rather that the predictor predictions superior for such years...**
No that was not the intended meaning for the sentence, so I have clarified it: "These results show that the combined prediction is closer to the statistical rather than the

dynamical prediction. In case the combined prediction is not the best one, it is in almost all cases better as one of the two."

So essentially, we have the following combinations (1 model, 2 predictor, 3 combined):
123: 1
132: 4
213: 0
231: 17
312: 4
321: 9
The statements points out that when 3 (combined) is not the leading prediction, then it is only once the worst one (so 123 and 213 are only selected once).

**2.6 Discussion and Conclusion lines 182-184: Do you have any idea, why, on the one hand, the combined model is at first place almost as often as the statistical Model, but on the other hand is much more often at the second place? Is it possible, that the statistical model may not capture certain special cases, which are then taken into account by the dynamical model? Maybe an investigation of the corresponding years would certainly be very revealing.**
We are working on the sub-selection algorithm with several scientist and students for more than five years now. We were not able to determine any clear argument in ways you intend. In the preparation for this manuscript over the last year I have tested many approaches, looked at autocorrelations and similar things, but there was nothing to report. Going into physical detail why single years are now especially preferring one over the others are certainly interesting Master-thesis topics, but are certainly far out of the scope of the manuscript. So no, there is certainly no simple answer, but there might certainly be aspects for the future someone might be interested to look at.

[Figure]

**3 Technical Corrections line 17: the NAO describe → the NAO describes**
**line 19: (Hurrell (1995)) → (Hurrell, 1995)**
**line 21: (Folland et al. (2019)) → (Folland et al., 2019)**
**line 22: (Butler et al. (2016)) → (Butler et al., 2016)**
**line 32: sub-sampling → subsampling**
**line 85: (Rubner et al. (2001)) → (Rubner et al., 2001)**
**line 86: (Düsterhus and Hense (2012)) → (Düsterhus and Hense, 2012)**
**line 123: both other prediction → both other predictions**
**line 150: inflence → influence**
**line 170: three prediction → three predictions**
**line 200: (Thorarinsdottir et al. (2003)) → (Thorarinsdottir et al., 2003)**
**line 201: the approach allow → the approach allows**
**line 206: disadvantage ist → disadvantage is**

Corrected.

**4   Reviewer 3:**

**I agree with the general comments of both Reviewer 1 and Reviewer 2: the
methodology is relatively straightforward and the text is well structured and
easy to follow. While I found the use of the Earth Mover's Distance interesting,
I support Reviewer 1 comments on using the CRPS. The reason for this being
that if one is to introduce a new methodology, it is easier to demonstrate its
usefulness by using the tools of the trade that people are already familiar with.
Using such score will make it easier to compare with other post-processing
attempts and might improve the chances that the methodology will be adopted.**

[Figure]

I have explained in the other reviews my motivation and the background for the inclusion of the CRPS-like IQD.

**Also, I found the terminology used to refer to the different forecast quite confusing. For one, the author uses model to refer to the output of the dynamical forecast whereas really the 3 sets of predictions are actually models, albeit just different types of models. I would replace "model" with either dynamical model or the name of the forecast system (or something along those lines) throughout the text and in table 1. I would also suggest changing the name predictor model to something else, just to avoid the repetition with the word prediction (which comes up a lot) and to differentiate with the actual predictors (e.g. on line 147: "to the predictor prediction"). Maybe it can be changed to statistically-based, or something along those lines. And not refer to the combined model as a hybrid model or statistical-dynamical model, as in the introduction? Although referring to it as the combined model is ok and doesn't lead to confusion in this case.**

I have renamed the models as dynamical and statistical and stuck to the naming of a combined model.

**Introduction**
**I think the manuscript would benefit from a short paragraph on the skill of climate models at predicting the NAO. At the moment, there is only 1/2 a sentence on this. In particular, they should mention that Scaife et al. (2014) do find skill at predicting the winter NAO with a different forecast system.**
I have added context and cited as requested the named article. I added the introduction of "various" to the existing sentence to make sure that it is clear that it was not just

the MPI-ESM who has WNAO skill. The reference to Butler et al (2016) is covering various models (including (GLOSEA)) where some show more, some show less prediction skill. I also added the word "significant" that stresses that it is measured by correlation skill a real thing. Every more detail would lead to a numbers game. Those I see highly critical. While I can understand the importance of numbers from the view of a National Weather Service, which has to fulfil targets, the focus on numbers is in my personal opinion extremely harmful for the community and its scientific development.

**Methodology**
**I would mention the equivalent resolution of the atmosphere and ocean model in either km or degree. I would guess many readers won't know what T63 and/or T0.4 correspond to.**
I have added the km and degree equivalent.

**I would also specify the components used to initialize the model as well as the observational dataset used to verify the predictions.**
Added as wished.

**Section 3.4 It took me a while to understand what was meant by "To estimate the post-processed variable field", as all the discussion up until that point was about computing the NAO. I would add an introductory sentence to help the reader transition from the NAO to the meteorological variables we actually care about (tas, pr, ...).**

I have added a sentence for Reviewer 2, which includes the statement that WNAO influences important climate variables.

[Figure]

**And I'm with Referee 1 on equation 5: I don't quite understand what is happening here.**

I have clarified it as explained to Reviewer 1.

**Minor points Line 31: an dynamical -> a dynamical**
**Line 52-54: "The skill is ..." I don't understand this sentence. Please reformulate**

Corrected. Sentence split in two.
"As a general skill measure the community applies correlation skills. Those measures have shown in recent years significant hindcast skill for several different prediction systems (Buttler et al, 2016)."

**Line 70: " Each predictor predicts statistically..." That's a bit awkward. You should reformulate.**

Reformulated.
"Each predictor makes a prediction from the climatic state taken from the ERA interim reanalysis (Dee et al, 2011) before the initialisation of the dynamical model for a WNAO value in the following winter."

**Line 81" a clearer signal as -> a clearer signal than**
**Line 106: By this approach -> With this approach**
**Line 120: in contrast towards -> with respect to**

Corrected.

[Figure]

**Line 126: "but in case one of the two is better than the combined prediction, the latter beats in nearly all cases the one remaining". Isn't this what we would expect? If we are to average two models in a 3rd one and #1 is the best, I would expect that #3 would be #2.**

Glad that you see it this way. As the comments of the other reviewers show it is not that clear. With the experience of many review rounds on the subsampling algorithm, it was a constant argument whether the combined prediction is something new or one of the two (so basically whether the information of the dynamical model disappears completely within the combination process). Showing that it is a mix out of the two is therefore for those discussions an important step forward.

**Line 140: "are hardly existing". Please reformulate.**

Reformulated:
"The comparison to the statistical prediction shows only small differences."

**Line 143: "Towards the model prediction" I don't know what you mean here.**

Reformulated:
"Compared to the dynamical prediction..."

**Line 148: In comparison to the predictor prediction... significantly decreased in their hindcast skill." Please reformulate this sentence.**

[Figure]

Reformulated:
"In the comparison to the statistical prediction the combined prediction shows significantly lower hindcast skill at areas over Greenland and the British Isles."

**Line 159: "In the comparison between.. can be explained by the statistical model." I don't understand what the author is trying to say here.**

Reformulated:
"In the comparison between the statistical and the dynamical prediction the increases and decreases are consistent with what has been seen for the dynamical compared to the combined prediction. This consistency shows that the statistical model plays a dominating role in the combination."

**Line 181: "a too high weight -> too high of a weight Line 187: "We have seen that ... in different ways" Do you mean that they provide a different perspective?**

Corrected and Reformulated:
"We have seen that the two categories of scores show the hindcast skill of the different forecasts from a different perspective."

**Line 193: "Also, unlike ... in a consistent way". Please rephrase.**

Reformulated:
"Compared to the correlation the evaluation of the skill can be decomposed in a consistent way for every single year."

**Line 196: "As each year... To prevent this other references," I don't understand this.**

[Figure]

Reformulated:

"As each year is compared to the climatology, a value close to the climatology can have a huge influence by creating substantive negative scores. To prevent this, the application of other references, like uniform distributions over the whole measurement range, can be an appropriate alternative."

**Line 203: cannot anymore drive -> cannot drive Line 205: The used evaluation -> The evaluation**

Corrected.

**Figures I would combined Figures 2, 3 and 4 into Figure 2a, b, c. Also, I would put the x-axis range from -2.5 to 2.5.**

The figures created in a way that they will fit into a single column in the final edit, without an enormous caption. I have before submission tested both ways and having it in a single figure is overwhelming. Therefore I will stick with the three different figures. The x-axis is chosen because of displaying it in the correct mathematical sense, so the actual information going into the calculations. As I have chosen the support of the NAO value as 3 standard deviations, it would be a false statement, when I would limit the range artificially afterwards.

**The legend in most figure is quite difficult to read.**

I have increased the font size for the legends.

[Figure]

**Figures 2-5: I would remove the [ ] when there are not units.**

I am very concerned with this request, as this will lead to physically wrong figure descriptions. Nevertheless, I fulfilled the request.

**Figure 2: I would make the individual members more visible.**

I chose a darker gray.

**Figure 4: I would display the obs in black, as it is the reference.**

It would make it much harder to identify to use red for another prediction. Especially the figure for the dynamical prediction would be much harder to understand if I would use red here. As I have to take care for colour impaired viewers, red (as well as green) should have a unique behaviour, which is currently given. Therefore, I will not change the colour.

**Figure 5: I see no dark green in the figure, despite what the caption says. Also, I would pick different colors to differentiate between the different models: light blue and dark blue makes it difficult to see what's going on.**

Mistake on my side. It should have been pink all along as in the figures before. Had changed the colour scheme, without changing it for all figures.

**Figure 6: The font on the colorbar is way too small. Also, I would change the range of the color for columns 2 and 3. Figure 7: The font on the colorbar is also**

[Figure]

**way too small.**

Colourbar was changed and adapted for both plots. Unfortunately, changing the range in columns 2 and 3 in figure 6 does not help much. For the main column I have to use the range $-0.9 : 0.9$ to fit everything in. In column 2 I already have to choose $-0.6 : 0.6$ to achieve this. As column 2 and 3 should be the same colourbar it does not help much and the colours hardly change (especially for column 3). As the main argument is that column 2 is in order of column 1 and column 3 is close to be inexistent, I think the message is simplest transported by not meddling with the colourbar for different plots.

**I would also change the colorbar for Figure 7 to a monochrome as opposed to red/blue.Red/blue is better when we have 0 in mid-range, but it is not the case here.**
It is right that there is no 0 in the middle, but a white colour means that both predictions have won the comparison in about equal number of years. I omitted a shift in the values (so to a centred around 0 variable), to be consistent with the NAO prediction analysis. So by the colour it is immediately clear, which prediction is better, which would not be the case in a monochromic colour-space.
* * *

---

## Author Response (AR2)

**Seasonal statistical-dynamical prediction of the North Atlantic Oscillation by probabilistic post-processing and its evaluation**

André Düsterhus

January 10, 2020

**1 General**

Thanks a lot for these important suggestions. I answer them bellow point-by-point. I have also added the annotations in the plots again, which I had forgotten in the last resubmission.

**2 Editor**

**Comments to the Author: Dear André Düsterhus,**

**I agree with the positive reports of the three reviewers. In light of the reviewer's comments and the revised version of the manuscript, I would like to suggest a few additional minor revisions.**

**- page 2, line 35: I suggest to change "climate science" to "atmospheric science", "weather prediction" or similar as all references are from the NWP literature.**

Changed to "atmospheric science".

**- page 2, line 50: Is the model resolutions of 40 km over the ocean correct? I would have expected a lower resolution compared to the T63 model.**

Yes that is correct. Of course in the relevant regions there will never be a resolution this coarse, as the numerical North pole will be placed in a way to have a finer resolution where it matters (e. g. the North Atlantic). That is why these numbers of degree and km are so misleading and in my opinion we should stick generally only to the technical resolutions. Nevertheless, as many have no understanding of these terms it might be useful (and at the same time confusing) for some readers to include the equivalents.

**- page 3, line 64-65: The revised version of this sentence is more clear (see comments by Anonymous Referee #3), however, I suggest to replace "shown" by "indicated" or similar**

to clarify further.

Changed to "indicated".

**- Section 3.3: Personally, I was very happy to see proper divergence measures used for evaluation. However, as evident from all review reports, this concept seems to be not very well known in the community. Therefore, I suggest to make the difference between IQD and CRPS more clear, for example by extending the discussion in lines 111-112 on page 4 by stating more explicitly that the IQD extends the CRPS towards observations that are not deterministic, but "distributions".**

I clarified it further. Added the sentence: "As a consequence, while CRPS needs to have a point observation, the IQD can take into account the full uncertainty distribution of an observation."

**- page 5, line 119-121: The sentence starting with "EMD is a metric..." will be very hard to understand for readers not familiar with Thorarinsdottir et al. (2013). Perhaps it can be rephrased in a more generic way without referring to the concepts of "k-proper" or "asymptotically proper"? Further, I am not sure that I understand the reasoning behind the statement in the subsequent sentence claiming that EMD specifically prefers underdispersed model simulations. Why does this follow from being only asymptotically proper, but not a k-proper divergence measure? In my view, this only implies that EMD may prefer incorrect models over the correct one, but may those incorrect models not as well be overdispersed?**

The argument have been taken from Thorarinsdottir et al. (2013) (THO13) and there are reasons why I omitted it in the original submissions. Generally, it makes the paper much more complex in the statistical sense, but as the reviewer have mentioned that seems to be ok. Taking only the IQD will not prevent me from explaining the problematics (as squaring is a very unintuitive choice for most), so I included both. I have tried to simplify the sentence to "While EMD is a metric measuring the distance between the pdf it is in contrast to IQD not a proper score." I would like to prevent to go into more detail here, because the difference between the two measures given by THO13 with "(iv) the divergence function ought to be "mathematically well behaved and well understood."" is quite hard to formulate in a correct and at the same time understandable way for non-verification-statisticians. Concerning the "underdispersed" statement, I have taken it directly from THO13 "..., thereby suggesting that $d_{AV}$ encourages underdispersed model simulations." While I see your questions as valid, it follows, as far as I understand the paper, out of MC-simulations (p 528 bottom). In this part of the manuscript, I just point to THO13 and their mathematical argumentation and do not see any argument, which contradicts their findings (and it would be overstretching this article to redo the mathematical background entirely). As I made clear in the manuscript that the IQD is to be preferred out of theoretical argumentations (even when EMD is much more intuitive), I think it is a fair way to handle it. Also as I stated in my last reply: I think it is important to get theoretical results out of the mathematical literature step by step into the more applied literature to make it more approachable for everybody. At the IMSC in Toulouse last year it was apparent in the Verification session that handling observations with uncertainties is an important next step. So this manuscript should show that choices matter, and makeing this way reders aware of the problematics we face when we approach this new horizon. Showing both metrics and their differences in

this manuscript is therefore in my view a valid approach to achieve this aim.

**- page 9, line 237-238: For evaluating point forecasts the RMSE can be used without an (implicit or explicit) assumption of Gaussianity, see, for example Gneiting (2011, https://doi.org/10.1198/jasa The statement in the sentence starting with "An important advantage ..." thus seems a bit misleading and I suggest to rephrase accordingly.**

I have taken the sentence out, as it is unnecessary at this point of the storyline. The problem with non-normal distributions of variables for correlations is discussed some paragraphs below.

 **- page 17: The caption of Figure 5 mentions a "dark green" color which is not present. Should this read "red" instead?**

Is changed.

 **Best regards, Sebastian Lerch**

---

## Author Response (AR3)

**Seasonal statistical-dynamical prediction of the North Atlantic Oscillation by probabilistic post-processing and its evaluation**

André Düsterhus

January 16, 2020

**1 General**

Thanks for the quick decision and the raised points. Both were clearly a fault on my side.

**2 Editor**

**Comments to the Author:**
**Thank you for the clairifications and the swift response.**

**I only have two very minor remaining comments regarding the newly added sentence starting on page 5, line 121 ("While EMD is a metric measuring..." ): "pdf" should probably read "pdfs" instead. Further, I would like to suggest to replace the term "proper score" by "proper metric", "proper divergence measure" or similar to further clarify the difference in application of proper scores (such as the CRPS) and proper divergence measures (such as IQD).**

All changes applied as suggested.